# Exploring Silicon Isotope Fractionation by Silicoflagellates: Results from a KOSMOS Experiment off Peru

Patricia Grasse<sup>1,2,\*</sup>, Kristin Doering<sup>3</sup>, Allanah J. Paul<sup>2</sup>, Avy Bernales<sup>4</sup>, Sonia Sanchez Ramirez<sup>4</sup>, Elisabeth von der Esch<sup>5</sup>, Michelle Graco<sup>4</sup>, Tim Boxhammer<sup>2</sup>, Lennart T. Bach<sup>6</sup>, Ulf Riebesell<sup>2</sup>, Martin Frank<sup>2</sup>

<sup>1</sup>German Centre for Integrative Biodiversity Research (iDiv) Halle-Jena-Leipzig, Leipzig ,04103, Germany

<sup>2</sup>GEOMAR Helmholtz Centre for Ocean Research Kiel, 24148 Kiel, Germany
 <sup>3</sup>Department of Geoscience, Aarhus University, 8000 Aarhus, Denmark
 <sup>4</sup>Dirección General de Investigaciones Oceanográficas y Cambio Climático, Instituto del Mar del Perú (IMARPE), Callao, Peru
 <sup>5</sup>Institute of Hydrochemistry, Chair of Analytical Chemistry and Water Chemistry, Technical

15 University of Munich, Munich, Germany

<sup>6</sup>Institute for Marine and Antarctic Studies, University of Tasmania, Hobart, Tasmania, Australia

\*Correspondence to: Patricia Grasse (patricia.grasse@idiv.de)

#### Abstract.

20 The Peruvian Upwelling is known for its exceptionally high surface water productivity and the presence of one of the world's largest Oxygen Minimum Zones. The upwelling of silicate-rich subsurface waters typically supports diatom-dominated primary productivity in this region. However, warmer surface waters and subsequent changes in stratification and nutrient supply can cause a shift in plankton communities from diatoms to dinoflagellates and silicoflagellates, which affects the silicon (Si) and carbon (C) cycles.

carbon (C) cycles. In 2017, we investigated the Si cycle in a field experiment off the coast of Peru. Pelagic mesocosms (~55,000 L) were deployed for 50 days from February to April to simulate upwelling conditions, which coincided with a coastal El Niño. This unique setting allowed us to study the evolution of stable silicon isotopes in seawater ( $\delta^{30}Si_{dSi}$ ) and its direct comparison to the produced biogenic material ( $\delta^{30}Si_{bSi}$ ) without the influence of unaccountable water mass mixing. On day 12, approximately 40% of the surface water of the mesocosms was replenished with nitrate-depleted deep water (low N:Si and N:P ratios), which strongly influenced the phytoplankton community. Prior to the addition of the deep water, the phytoplankton community was dominated by diatoms but shifted towards a pronounced dominance of flagellates, including silicoflagellates. At the beginning of the experiment, when diatoms dominated 35 the phytoplankton community, the  $\delta^{30}$ Si<sub>dSi</sub> distribution in the surface water (+1.4 % to +2.5 %) was within the same range as observed in previous seawater studies in the Peruvian upwelling. After deep water addition, low N:Si (0.02 to 0.2 mol/mol), strongly deviating from the preferred 1:1 ratios for diatoms, favored silicoflagellate (and dinoflagellate) growth and resulted in higher  $\delta^{30}$ Si<sub>dsi</sub> values (up to +4.1%) in the surface waters. The strong increase in  $\delta^{30}$ Si<sub>dSi</sub> was associated with low  $\delta^{30}$ Si<sub>dSi</sub> values 40 (-0.26 to +0.65 %) caused by high fractionation factors of stable silicon isotopes between seawater and silicoflagellates. For the first time, the field experiment allowed us to determine the Si isotope fractionation factor for silicoflagellates ( $\varepsilon^{30}_{\text{silico}} = -3.63$  %), which is remarkably high compared to diatoms (-1.1 ‰) and offers a novel tool to study changes in the present and past marine silicon cycle.

## 45 1 Introduction

The Peruvian Upwelling is characterized by exceptionally high rates of productivity induced by Ekman suction of subsurface waters enriched in dissolved silicic acid (Si(OH)<sub>4</sub>, hereafter referred to as dSi) and other macronutrients such as phosphate and nitrate (e.g., Bruland et al., 2005; Strub et al., 1987). The presence of nutrient-rich surface waters and high solar radiation at this low latitude results in high primary productivity along the coast throughout the year, with chlorophyll (Chl *a*) reaching concentrations of up to 10 mg m<sup>-3</sup> (Echevin et al., 2008; Franz et al., 2012). The export of plankton and organic matter from the surface and its decomposition at depth produces one of the largest subsurface oxygen minimum zones (OMZs) in the global ocean (Karstensen et al., 2008; Pennington et al., 2006). Fluctuations in phytoplankton biomass have been linked to intra- and interannual environmental variability, directly affecting the nutrient stoichiometry and phytoplankton communities (Echevin et al.,

https://doi.org/10.5194/egusphere-2025-5079 Preprint. Discussion started: 28 October 2025 © Author(s) 2025. CC BY 4.0 License.

2008; Ochoa et al., 2010). Coastal upwelling off Peru occurs year-round, reaching peak intensity during the austral winter (June–September) when strengthened trade winds drive nutrient-rich conditions, marked by high concentrations of dSi in surface waters. This environment is highly favorable for the growth of diatoms, a group of phytoplankton that have a strict requirement for silica, which they use to build their cell walls, known as frustules (e.g., Ochoa et al., 2010; Tréguer et al., 2018; Werner, 1977). Diatoms play a crucial role in the marine silicon and carbon cycle, as they are the primary organisms responsible for the uptake of dSi and carbon from seawater (Ragueneau et al., 2000).

In contrast, during the austral summer (December–March) or during El Niño–Southern Oscillation (ENSO) warm phases, coastal upwelling is weakened or disrupted. This results in warmer sea surface temperatures, enhanced stratification, and a reduction of dissolved silicate (dSi) concentrations in surface waters. As a result, diatoms, which are highly dependent on dSi, become less competitive, leading to shifts in the phytoplankton community. Later successional phytoplankton stages, such as dinoflagellates and silicoflagellates, which have different ecological and nutritional requirements, become more dominant (e.g. Ochoa et al., 2010). These shifts in phytoplankton composition strongly affect the marine silicon cycle, but also the carbon cycle dinoflagellates and silicoflagellates have lower silica requirements and different carbon export efficiencies compared to diatoms.

With regard to the marine silicon cycle, diatoms are the most important phytoplankton group in the Peruvian Upwelling system, whereas other silicifying organisms, such as radiolaria and silicoflagellates, play a minor role due to low abundances. These organisms contribute less significantly to the silicon cycle and are generally absent in surface waters during periods of strong upwelling, when diatom blooms are most prevalent (Franz et al., 2012; Grasse et al., 2021).

Several studies in the Peruvian Upwelling investigated the stable Si isotope distribution in seawater (δ³0Si<sub>dsi</sub>) and biogenic particles (δ³0Si<sub>bsi</sub>) to improve our understanding of the Si cycle at present (Ehlert et al., 2012; Grasse et al., 2013; 2016; 2020; 2021) and in the past (e.g., Doering et al., 2016a; 2016b; Doering et al., 2019; Ehlert et al., 2013). The main factor controlling δ³0Si in the euphotic zone is dSi utilization by siliceous phytoplankton, mainly diatoms, which preferentially incorporate the lighter Si isotopes into their frustules thereby elevating the δ³0Si of the surrounding seawater (e.g., De La Rocha et al., 1997). Several culture studies have shown that the fractionation factor between seawater and diatoms (³0ε<sub>diatom</sub>: mean -1.1 ± 0.4 ‰, 1 s.d.) is species-dependent (De La Rocha et al., 1997; Sutton et al., 2013) and appears to be independent of temperature (12 to 23°C; De La Rocha et al., 1997), pCO<sub>2</sub> concentrations (Milligan et al., 2004), and growth rate (Sun et al., 2014). However, some of the results of previous studies are contradictory, and Meyerink et al. (2017) have suggested that nutrient availability (e.g., dSi, Fe, NO<sub>3</sub>) affecting growth rate, Si uptake, and bSi content in the cell influences Si isotope fractionation. In addition, isotopic fractionation may be related to the biochemical pathways involved in Si metabolism and may reflect the organism's affinity for dSi and efficiency of Si uptake and utilization.

110

In terms of the fractionation factor, the best-studied group is diatoms, but other marine silicifiers, such as radiolaria, sponges, and choanoflagellates, also show discrimination of the heavy Si isotopes during incorporation. While radiolaria appear to have Si isotope fractionation factors similar to diatoms (Doering et al., 2021), the highest fractionation has been observed for sponges and choanoflagellates (up to -5 \infty and -7 \infty, respectively; Hendry & Robinson, 2012; Marron et al., 2019; Sutton et al., 2018). The fractionation factor for silicoflagellates ( $^{30}\epsilon_{Silico}$ ) is still unknown. It has remained unclear what exactly the controlling factors of fractionation are and why differences in the fractionation factors are observed between different phylogenetic taxa and even on a species level.

The assessment of a fractionation factor between marine siliceous organisms (e.g., diatoms) and seawater ( $\varepsilon^{30}$ Si) during the field study can be based on conceptual models either assuming a Rayleigh-type (closed system) system or a steady-state model (open system; see Grasse et al., 2021). Applying these models to a highly dynamic system such as the Peruvian Upwelling is challenging, since 105 the temporal evolution of  $\delta^{30}Si_{dSi}$  is often oversimplified. A more accurate reconstruction of dSi utilization requires accounting for additional processes, including admixture from multiple sources (vertical and horizontal nutrient supply, as well as repeated nutrient intrusions), and the integration of dissolution processes (Grasse et al., 2021). In addition, the dissolution of biogenic material in the euphotic zone may bias estimations of the fractionation factor towards lower values (Grasse et al., 2013; 2016; 2021).

In 2017, we conducted a KOSMOS (Kiel Off-Shore mesocosms for Future Ocean Simulations) experiment off Lima (Peru) for 50 days to understand how upwelling of water masses with different nutrient stoichiometries (N:Si and N:P) influences pelagic biogeochemistry and plankton succession. The unique setting of the experiment made it possible to study the evolution of  $\delta^{30}Si_{dSi}$  and  $\delta^{30}Si_{dSi}$  in a 115 closed system without the influence of horizontal water mass mixing. We observed a shift from a diatom-dominated community (day1-10, phase I) to a (silico-)flagellate-dominated community (day 13 to 42, phase II) after the addition of deep water (DW), which allowed us to gain further insights into the Si cycle off Peru and to provide, for the first time, a fractionation factor for silicoflagellates.

## 120 **2 Methods**

## 2.1 Mesocosm set-up and sampling

Eight mesocosms (M1-M8) were deployed 6 km off the Peruvian Coast close to San Lorenzo Island (12.0555°S, 77.2348°W). Each mesocosm consisted of a cylindrical, 17 m long polyurethane bag (2 m diameter) attached to a conical sediment trap (2 m) in a 8m tall flotation frame. On 25th of February the 125 water was enclosed the polyurethane bag containing approximately 55 000 liters of water (Fig. 1). The experiment started on the 25th of February 2017 (Day 0) and was conducted for 50 days (for details on the experiment and sampling schedule, see Bach et al. 2020). After their closure, the waters inside the

mesocosms remained isolated from surrounding Pacific seawater. Repetitive sampling of the water column within the mesocosms included an integrated water sample from the mixed layer (ML) and the bottom layer (BL). The lower sampling depth for the ML was adjusted during the experiment to account for a shift in the ML and oxycline depth and included 5 m, 10 m, and 13 m, respectively (Table S1).

Fig. 1: (a) Schematic figure of a KOSMOS (modified from Bach et al., 2020). Integrated seawater samples were taken from the Mixed Layer (ML) and the Bottom layer (BL). Please note that the ML/BL sampling depth was adjusted in the course of the entire experiment. This study only discusses data from the ML. Further details are provided in section 2.1. See Bach et al. (2020) for a detailed location map and experimental setup.

DW for the simulated upwelling event was collected along the Instituto del Mar del Perú (IMARPE) time-series transect (Graco et al., 2017). The water was sampled on Day 5 (St. 1; 30m, St. 12.028°S, 77.22°W) and Day 10 (St.3, 70m, 12.04°S, 77.38°W). Samples from both stations were characterized by different nutrient ratios and therefore described as "extreme" (DIN:P of 0.2 mol/mol; DIN:Si of 0.02 mol/mol) and "moderate" (DIN:P of 1.7 mol/mol; DIN:Si of 0.2 mol/mol) DW. On the 8th and 9th of March 2017 (Days 11 and 12), we exchanged water enclosed in each mesocosm with water collected from station 1 (M1, M4, M5, M8) or station 3 (M2, M3, M6, M7). The exchange was carried out in two

https://doi.org/10.5194/egusphere-2025-5079 Preprint. Discussion started: 28 October 2025 © Author(s) 2025. CC BY 4.0 License.

steps using a submersible pump (Grundfos SP 17-5R, pump rate ~ 18 m³ h¹). On Day 8, BW was exchanged. We installed the pump for about 30–40 min in each mesocosm and pumped 9 m³ out of each bag from a depth of 11 to 12 m. On Day 11, the pump was installed inside the collector bags, and 10 m³ of water were injected into a depth of 14 to 17 m depth. On Day 12, the procedure was repeated to exchange the surface water. In that case, approximately 10 m³ of water were removed from 8 to 9 m depth and replaced with 12 m³ water evenly to the depth range from 1 to 9 m. This corresponded to an addition of 40% DW to the surface (mixed) layer.

Sampling and CTD casts within the mesocosms were undertaken from small boats that departed from La Punta harbor (Callao) and were transported to laboratories in Club Náutico Del Centro Naval and the Instituto del Mar del Perú (IMARPE) for filtration and nutrient measurements. Every  $2^{nd}$  day, subsamples for nutrients and natural Si isotopes were sampled from the mixed layer (ML). For  $\delta^{30} \text{Si}_{dSi}$  and  $\delta^{30} \text{Si}_{bSi}$ , 115 to 2000 ml of seawater was filtered through a 0.65  $\mu$ m Polycarbonate filter (Whatman®, 0.65  $\mu$ m pore size, 47 mm). Immediately after filtration, the seawater filtrate for  $\delta^{30} \text{Si}_{dSi}$  was acidified to pH 2 and stored in the dark. Filters for  $\delta^{30} \text{Si}_{bSi}$  were dried at 40°C. Dissolved and particulate Si samples were later processed at GEOMAR, Helmholtz Centre for Ocean Research, Kiel, Germany.

## 2.2 Dissolved inorganic nutrients and bSi concentrations

Samples for inorganic nutrients were filtered (0.45 μm filter, Sterivex, Merck) immediately after arrival in the laboratories of IMARPE. The subsequent analyses of dSi, PO<sub>3</sub><sup>4-</sup>, NO<sub>3</sub><sup>7</sup>, NO<sub>2</sub><sup>7</sup> concentrations were carried out using an autosampler (XY2 autosampler, SEAL Analytical) and a continuous flow analyzer (QuAAtroAutoAnalyzer, SEAL Analytical) connected to a fluorescence detector (FP-2020, JASCO). Phosphate and dSi were analyzed colorimetrically following the procedures by Murphy and Riley (1962) and Mullin and Riley (1955), respectively. Nitrate and nitrite were quantified via the formation of a pink azo dye as established by Morris and Riley (1963). Ammonium concentrations were determined fluorometrically (Kérouel & Aminot, 1997). The accuracy was monitored by including certified reference materials (CRM; BW, KANSO) during measurement sessions and ranged between ±5 % to ±10 %. For further analytical details see Bach et al. (2020). BSi filters were leached with 0.1 M NaOH at 85°C in 60 mL Nalgene polypropylene bottles at Club Náutico Del Centro Naval. After 135 minutes, the leaching process was terminated with 0.05 M H<sub>2</sub>SO<sub>4</sub>, and the dSi concentration was measured spectrophotometrically following Hansen & Koroleff (1999). The bSi concentration was measured for all size fractions, as well as for size classes smaller than 20 μm and larger than 20 μm.

#### 2.3 Phytoplankton assemblages

Seawater subsamples from the ML (25 ml and 50 ml) were analysed for phytoplankton assemblages (also including dead cells) at the laboratories of IMARPE (Instituto del Mar del Perú, La Punta). Cell counts were carried out according to the method of Utermöhl (1958), in which the sample was allowed

to settle for approximately 24 hrs. Cells were counted under Nikon and Leica inverted light microscope at at ×125 magnification (10 x = 10.01 µm; ocular: 12.5 x) and expressed in cells L<sup>-1</sup> x 10<sup>6</sup>. In addition, we calculated the relative abundance (in %) considering only marine silicifiers (diatoms and silicoflagellates) and the relative abundance (in %) for siliceous and non-siliceous plankton (e.g. dinoflagellates, coccolithophores) from microscopy data. More information on all phytoplankton groups analyzed, including pigment data, can be found in Bach et al. (2020).

#### 190 **2.4 Biovolume**

To estimate the biovolume of various species, we referred to the 2024 Nordic Microalgae Biovolume List (NOMP, Olenina et al., 2006). This resource is among the most comprehensive compilations currently available. Table S2 presents examples of commonly encountered diatom species (e.g., *Skeletonema costatum, mean diameter of 6 μm*), some of the largest observed species (e.g., *Coscinodiscus wailesii, mean diameter 252 μm*), as well as silicoflagellates (mean diameter between 20 and 30 μm). For *Dictyocha octonaria*, the biovolume and diameter could not be directly retrieved; instead, the diameter was determined from SEM images (example shown in Fig. 5d), and the biovolume was estimated by assuming a half-sphere shape, which is also applied for *Dictyocha fibula* (Olenina et al., 2006).

$$(1) V = \frac{\pi}{12} * d^3$$

Where V is the Biovolume, and d the diameter. Given the broad variability in biovolume, the values provided should be considered as approximate estimates.

## 2.5 Sample preparation and measurements of $\delta^{30}Si_{dSi}$ and $\delta^{30}Si_{bSi}$

- For preparation of samples for δ<sup>30</sup>Si<sub>dSi</sub> measurements, the seawater pH (9 to 10) was raised with 1 M NaOH to scavenge dSi with the precipitated Mg(OH)<sub>2</sub> (MAGIC, Karl & Tien, 1992; Reynolds et al., 2006, Grasse et al., 2016). BSi filters for δ<sup>30</sup>Si<sub>bSi</sub> measurements were treated according to Varela et al. (2004) including a leaching step with 0.2 N NaOH in a 95°C water bath (for details, see also Grasse et al., 2021).
- The dissolved sample (dSi, bSi) was loaded onto a cation exchange column (AG50X8, 200 mesh, Biorad®; for details, see Grasse et al. (2021). Samples with low dSi concentrations ( $<4 \,\mu mol \, L^{-1}$ ) were gently evaporated to double the concentration after column chemistry. According to Hughes et al. (2011), this should not affect  $\delta^{30}$ Si. Detailed preparation protocols are described in Ehlert et al. (2012) and Grasse et al. (2013).
- Samples were analyzed using an Aridus II nebulizer coupled to a Nu Plasma MC-ICP-MS (Nu Instruments<sup>TM</sup>, Wrexham, UK). Each analysis involved 50 to 60 cycles in sample-standard bracketing mode against NBS28. Si isotope compositions are reported in the δ notation, representing the deviation

245

of the isotope ratio of the sample ( $R_{sample}$ ) from that of a reference standard ( $R_{standard}$ ) in parts per thousand (‰).

$$\delta^{30}Si = \left(\left(\frac{R_{sample}}{R_{standard}}\right) - 1\right) * 1000$$

The accuracy of measurements was checked daily using solid reference standards and seawater standards. Repeated measurements of BB, Diatomite, and IRMM18 resulted in mean  $\delta^{30}$ Si of -10.66  $\pm$  018 ‰ (2 s.d., n = 14), +1.22  $\pm$  0.14 ‰ (2 s.d., n = 28) and -1.43  $\pm$  0.23 ‰ (2 s.d., n = 10), which are in good agreement with Reynolds et al., 2007. The seawater inter-calibration standards Aloha 1000 m and Aloha 300 m resulted in +1.25  $\pm$  0.16 ‰ (2 s.d; n = 53) and +1.72  $\pm$  0.10‰ (M2 s.d., n = 8), which is in excellent agreement with the mean values obtained by the GEOTRACES Si isotope inter-calibration study (+1.24  $\pm$  0.20‰; +1.68  $\pm$  0.35‰, Mean  $\pm$  2 s.d.; Grasse et al., 2017).

### 2.6 Calculation of the in situ fractionation factor for silicoflagellates

The mesocosms are closed containers, in which dSi was replenished on day 11 (bottom water) and day 12 (surface water). The fractionation factor between silicifiers and seawater (<sup>30</sup>ε<sub>silicos</sub>) can, therefore, be calculated after dSi replenishment, assuming a Rayleigh-type fractionation (closed system) (Mariotti et al. 1981; De La Rocha et al., 1997; Sun et al., 2014). Unfortunately, we were not able to calculate the fractionation factor prior to the addition of deep water (phase I), given that the dSi concentrations were partly increasing over time as a result of mixing and changes in the ML depth (see Fig. 2a). The Rayleigh-type model is expressed by following equations.

(3) 
$$\delta^{30}Si_{dSi} = \delta^{30}Si_{init} + {}^{30}\varepsilon_{mixed} \times ln(f)$$
  
(4)  $\delta^{30}Si_{bSi} = \delta^{30}Si_{init} - {}^{30}\varepsilon_{mixed} \times \left[\frac{f}{1-f}\right] \times ln(f)$   
(5)  $f = ([dSi]_{final}/[dSi]_{source})$ 

where  $\delta^{30}Si_{dSi}$ ,  $\delta^{30}Si_{init}$  and  $\delta^{30}Si_{bSi}$  are the Si isotope values of the substrate (dSi) and the initial dSi concentration on day 13 (no sampling on day 12) as well as of the product (bSi) on day 17, respectively. The remaining fraction f was calculated according to equation (5) using the dSi concentration from day 13 (dSi<sub>source</sub>) and 17 (dSi<sub>final</sub>). The time range between days 13 and day 17 was used as it marked the initial decline in dSi following the increase directly linked to deep water addition. After day 17, dSi was increasing again (Fig. 2a).

To calculate the fractionation factors in the three different mesocosms, equation (4) was solved for  $^{30}$  $\varepsilon_{\text{mixed}}$  and calculated according to the following equation (6) (Sun et al., 2014), which uses a Rayleightype model and takes into account the utilized fraction f.

$$(6)^{30} \varepsilon_{mixed} = -(\delta^{30} Si_{bSi} - \delta^{30} Si_{init}) / \left\{ \left[ \frac{f}{1-f} \right] \times \ln(f) \right\}$$

Please note that the calculated  ${}^{30}\varepsilon_{\text{mixed}}$  is a mixed Si isotope fractionation factor for silicifiers (diatoms and silicoflagellates). Using the relative abundances of diatoms ( $f_{\text{diatoms}}$ ) and silicoflagellates (1- $f_{\text{diatoms}}$ ),

we estimated the fractionation factor for silicoflagellates (<sup>30</sup>ε<sub>silicos</sub>) assuming a fractionation factor for diatoms (<sup>30</sup>ε<sub>diatoms</sub>) of -1.1‰. In addition, it was tested for the minimum and maximum fractionation factors reported for different diatom species (-0.5 to -2.1 ‰; Sutton et al., 2018).

$$(7)^{30} \varepsilon_{silicos} = (f_{diatoms} \times {}^{30} \varepsilon_{diatoms} + {}^{30} \varepsilon_{mixed})/(1 - f_{diatoms})$$

The offset ( $\Delta^{30}$ Si, also called the apparent fractionation factor) between  $\delta^{30}$ Si<sub>bSi</sub> and  $\delta^{30}$ Si<sub>dSi</sub> was calculated according to equation (8) (Table 1). The apparent fractionation factor is often determined in field studies investigating the marine Si cycle (e.g., Varela et al., 2019; Grasse et al., 2021)

(8)
$$\Delta^{30}$$
Si =  $\delta^{30}$ Si<sub>bSi</sub> -  $\delta^{30}$ Si<sub>dSi</sub>

The propagated error for  $\Delta^{30}$ Si is defined as follows

(9) 
$$\sqrt{2 s. d._{bSi^2} + 2 s. d._{dSi^2}}$$

### 2.7 Statistical analysis

Monto Carlo simulations were performed to propagate the uncertainty in the calculated <sup>30</sup>ε derived from analytical errors of Si isotope measurements (Robert & Casella 2004). 2000 Monte Carlo simulations were performed assuming a normal distribution. This allows a random generation of possible results that substitute a range of values based on a probability distribution of measured δ<sup>30</sup>Si values. The simulation was performed in Excel with the Analysis ToolPak, allowing random number generation within the 2 s.d. error.

#### 3 Results

The dissolved and particulate (in-)organic nutrient data have been described and discussed in detail by Bach et al. (2020). Here, we only list the most relevant findings for the investigations of the Si cycle.

The water column, enclosed at the beginning of the study, was thermally stratified with a thermocline at approximately 5 m, which shifted to approximately 10 m after DW addition. The thermocline roughly corresponded to the oxycline. Below, the water in the BL was depleted in oxygen concentrations (<50 μmol L<sup>-1</sup>) compared to the ML (>200 μmol L<sup>-1</sup>, Bach et al., 2020). δ<sup>30</sup>Si<sub>dSi</sub> was only analyzed in the surface waters from three mesocosms (M1, M2, and M7) as the measurements are highly time- and labor-intensive. These mesocosms were selected because they represented a large range in dSi concentrations (Fig. 2a). δ<sup>30</sup>Si<sub>bSi</sub> measurements were only conducted on selected filters for the estimation of the fractionation factor (Table 1).

## 3.1 Deep Water

Nutrient concentrations, as well as δ<sup>30</sup>Si<sub>dSi</sub>, were determined for both DW types ("extreme" and "moderate", see Table 1). Both DWs showed similar dSi concentrations (DW<sub>extreme</sub>: 17.4 μmol L<sup>-1</sup>; DW<sub>moderate</sub>: 19.6 μmol L<sup>-1</sup>) and PO<sub>4</sub><sup>3-</sup> concentrations (~2.5 μmol L<sup>-1</sup>) but deviated in their N:Si ratio

(0.02 versus 0.2) as well as N:P ratios (0.1 versus 1.7). The  $\delta^{30}\text{Si}_{dSi}$  values for DW "moderate" and DW "extreme" were indistinguishable within error with  $+1.54\pm0.12$  % and  $+1.47\pm0.19$  %, respectively.

Table 1: Nutrient concentrations,  $\delta^{30}Si_{dSi}$  and  $\delta^{30}Si_{bSi}$  for selected days in surface waters and the two different deep water (DW) types added to the ML on day 12. The colour code identifies the "very low N/P" (extreme) treatment (red) and the "low N/P" (moderate) treatment (blue). \*NO<sub>x</sub> =NO<sub>3</sub> + NO<sub>2</sub> · \*\*DIN=NO<sub>x</sub>+NH<sub>4</sub>. \*\*\* Values are used to calculate the fractionation factors. A detailed list of all the supplement Table SI. The experiment was

nutrients and stable isotope measurements in ML is in the supplement Table S1. The experiment was separated into three different phases, indicated in the second row.

|           |    |          | dSi          | NOx*     | P        | NH4      | BSi      | DiN:Si** | DIN/DIP** | Silicos        | Diatoms        | δ <sup>30</sup> Si<br>dSi | 2 s.d. | δ <sup>30</sup> Si<br>bSi | 2 s.d. | Δ <sup>30</sup> Si |
|-----------|----|----------|--------------|----------|----------|----------|----------|----------|-----------|----------------|----------------|---------------------------|--------|---------------------------|--------|--------------------|
|           |    |          | μmol L-1     | μmol L-1 | μmol L-1 | μmol L-1 | μmol L-1 | M/M      | M/M       | %              | %              | ‰                         | ‰      | ‰                         | ‰      | ‰                  |
|           |    | M1       | 2.13         | 0.14     | 0.70     | 0.00     | 2.64     | 0.07     | 0.20      | 0.01           | 99.99          | 1.42                      | 0.01   | 0.73                      | 0.16   | -0.69              |
| Day 1     |    | M2       | 3.95         | 1.77     | 0.98     | 0.47     | 2.37     | 0.57     | 2.29      | 0.17           | 99.83          | 2.22                      | 0.14   | 2.01                      | 0.07   | -0.20              |
|           |    | M7       | 4.11         | 2.63     | 1.03     | 4.39     | 4.01     | 1.71     | 6.82      | 0.01           | 99.99          | 2.27                      | 0.17   | 0.36                      | 0.14   | -1.91              |
|           |    | M1       | 2.21         | 2.28     | 1.26     | 0.91     | 2.52     | 1.44     | 2.53      | NaN            | NaN            | 2.29                      | 0.11   | NaN                       | NaN    | NaN                |
| Day 8     | I  | M2       | 2.25         | 1.46     | 1.39     | 0.62     | 1.99     | 0.92     | 1.50      | 0.29           | 99.71          | 2.49                      | 0.12   | NaN                       | NaN    | NaN                |
|           |    | M7       | 2.38         | 1.57     | 1.43     | 0.99     | 2.01     | 1.08     | 1.79      | 29.83          | 70.17          | NaN                       | NaN    | NaN                       | NaN    | NaN                |
| Day 10    |    | M1       | 2.95         | 0.00     | 1.35     | 1.26     | 0.59     | 0.43     | 0.94      | 1.17           | 98.83          | 2.08                      | 0.05   | 1.49                      | 0.07   | -0.59              |
|           |    | M2       | 2.76         | 0.00     | 1.40     | 0.79     | 0.69     | 0.29     | 0.56      | 1.99           | 98.01          | 2.31                      | 0.06   | 1.31                      | 0.27   | -1.00              |
|           |    | M7       | 2.93         | 1.68     | 1.54     | 1.50     | 0.54     | 1.09     | 2.06      | 96.72          | 3.28           | 2.07                      | 0.25   | 0.67                      | 0.20   | -1.40              |
| DW        |    | M1       | 17.44        | 0.30     | 2.56     | 0.25     | 1.98     | 0.02     | 0.21      | NaN            | NaN            | 1.47                      | 0.19   | NaN                       | NaN    | NaN                |
| addition  |    | M2; M7   | 19.56        | 3.96     | 2.49     | 0.30     | 1.17     | 0.20     | 1.71      | NaN            | NaN            | 1.54                      | 0.12   | NaN                       | NaN    | NaN                |
|           |    | M1       | 8.91         | 1.45     | 1.87     | 0.33     | 1.89     | 0.20     | 0.95      | 2.31           | 97.69          | 2.12                      | 0.25   | NaN                       | NaN    | NaN                |
| Day 13*** |    | M2       | 8.90         | 3.05     | 1.91     | 0.50     | 3.27     | 0.40     | 1.86      | 92.35          | 7.65           | 2.93                      | 0.21   | 0.02                      | 0.13   | -2.91              |
|           |    | M7       | 5.97         | 2.75     | 1.88     | 1.26     | 6.30     | 0.67     | 2.13      | 97.31          | 2.69           | 3.40                      | 0.15   | NaN                       | NaN    | NaN                |
| Day 17*** |    | M1       | 7.65         | 0.10     | 1.70     | 0.19     | 1.34     | 0.04     | 0.17      | 62.26          | 37.74          | 2.95                      | 0.19   | -0.26                     | 0.17   | -3.22              |
|           |    | M2       | 5.91         | 0.10     | 1.67     | 0.12     | 3.26     | 0.04     | 0.13      | 98.29          | 1.71           | 3.04                      | 0.11   | 0.12                      | 0.16   | -2.92              |
|           |    | M7       | 3.56         | 0.16     | 1.66     | 0.56     | 3.39     | 0.20     | 0.43      | 97.31          | 2.69           | 3.21                      | 0.13   | 0.65                      | 0.16   | -2.56              |
| Day 20    | ** | M1       | 7.52         | 0.02     | 1.59     | 0.00     | 0.68     | 0.00     | 0.01      | 76.03          | 23.97          | 2.33                      | 0.21   | 0.62                      | 0.12   | -1.71              |
|           | П  | M2       | 5.81         | 0.05     | 1.44     | 0.00     | 1.53     | 0.01     | 0.04      | 99.85          | 0.15           | 2.99                      | 0.02   | 0.41                      | 0.16   | -2.59              |
|           |    | M7       | 4.08         | 0.04     | 1.51     | 0.00     | 1.08     | 0.01     | 0.03      | 43.33          | 56.67          | 3.03                      | 0.14   | NaN<br>1.37               | 0.00   | NaN<br>0.75        |
| D 24      |    | M1<br>M2 | 7.66         | 0.03     | 7.66     | 3.79     | 0.45     | 0.50     | 0.50      | 40.00<br>89.87 | 60.00          | 2.12                      | 0.16   | 1.05                      | 0.00   | -0.75<br>-1.66     |
| Day 24    |    | M2<br>M7 | 6.18<br>4.84 | 0.02     | 1.51     | 0.04     | 0.52     | 0.01     | 0.04      | 75.00          | 10.13<br>25.00 | 2.78                      | 0.21   | 0.89                      | 0.27   | -1.89              |
|           |    | M1       | 6.39         | 0.01     | 1.50     | 0.03     | 0.80     | 0.01     | 0.02      | 0.00           | 100.00         | 1.55                      | 0.17   | 1.69                      | 0.07   | 0.14               |
| Day 38    |    | M2       | 6.58         | 0.04     | 1.59     | 0.19     | 0.73     | 0.04     | 0.13      | 44.44          | 55.56          | 2.46                      | 0.17   | 1.64                      | 0.06   | -0.83              |
| Day 30    |    | M7       | 5.13         | 0.03     | 1.37     | 0.11     | 0.73     | 0.02     | 0.09      | 0.00           | 100.00         | NaN                       | NaN    | 1.04                      | 0.20   | NaN                |
|           |    | M1       | 2.61         | 0.02     | 1.20     | 0.11     | 1.47     | 0.03     | 0.09      | NaN            | NaN            | 2.05                      | 0.17   | NaN                       | NaN    | NaN                |
| Day 44    |    | M2       | 4.27         | 0.00     | 1.24     | 0.38     | 3.77     | 0.09     | 0.31      | NaN            | NaN            | 2.34                      | 0.17   | NaN                       | NaN    | NaN                |
| 2m,       |    | M7       | 4.31         | 0.11     | 1.35     | 0.37     | 0.74     | 0.11     | 0.36      | NaN            | NaN            | NaN                       | NaN    | NaN                       | NaN    | NaN                |
|           | Ш  | M1       | 2.81         | 0.03     | 1.60     | 0.25     | 2.10     | 0.10     | 0.17      | 0.00           | 100.00         | 1.14                      | 0.19   | 1.84                      | 0.01   | 0.70               |
| Day 50    |    | M2       | 3.73         | 0.14     | 1.27     | 0.20     | 1.23     | 0.09     | 0.27      | 1.37           | 98.63          | 0.85                      | 0.28   | 0.98                      | 0.14   | 0.13               |
| 2, 55     |    | M7       | 3.50         | 0.14     | 1.42     | 0.12     | 0.91     | 0.07     | 0.18      | NaN            | NaN            | 2.38                      | 0.31   | NaN                       | NaN    | NaN                |

## 3.2 Dissolved nutrients and nutrient ratios in the euphotic surface layer

The dSi concentration in the ML initially ranged between 2.1 and 4.1 μmol L<sup>-1</sup> within the first ten days (phase I) in M1, M2 and M7 (Fig. 2a). This variability in dSi concentrations during the first ten days was mainly the result of a shift in sampling depth on day 4 to follow the deepening of the ML depth (Fig. 2a). This resulted in an increase in dSi compared to day 1 (Fig. 2b). After DW addition (Day 12), dSi in the ML increased to 9 μmol L<sup>-1</sup> (day 13) in M1 and M2 and 6 μmol L<sup>-1</sup> in M7, due to addition of DW with higher dSi (17.4 μmol L<sup>-1</sup>; DW: 19.6 μmol L<sup>-</sup>, Table 1).

Between day 13 and day 20, dSi in the ML decreased by 20 to 50% resulting in dSi concentrations between 3 and 8  $\mu$ mol L<sup>-1</sup> (Fig. 2 a,b). From day 20 to 35, dSi increased in M2 and M7, while it remained relatively constant in M1. By the end of the experiment (after day 35), dSi had

decreased to approximately 3  $\mu$ mol L<sup>-1</sup> in M1, M2, and M7, resulting in an overall drawdown between 310 30 and 70%.

During the first 10 days of the experiment DIN:dSi ratios (DIN=NO<sub>x</sub> + NH<sub>4</sub>) ranged between 0.5 and 1.5 mol/mol and decreased for both treatments after the DW addition (Fig. 2c). Between days 15 and 40, the DIN:dSi ratio was close to 0 and only slightly increased at the end of the experiment (after day 38) due to surface eutrophication with NH<sub>4</sub><sup>+</sup> by defecting seabirds (Inca tern, *Larosterna inca*, Bach et al., 2020, see also video from Boxhammer et al. (2019)).

DIN:P values showed a pattern similar to DIN:dSi. While they were higher during the first 10 days (1 to 7 mol/mol), they decreased sharply after the DW addition and were close to 0 mol/mol after day 15 with a slight increase towards the end of the experiment (Fig. 2d).

Fig. 2: Dissolved concentrations and nutrient ratios during the experiment. The deep water (DW) additions on days 11 and 12 are marked by a grey bar. M1 data are indicated with filled red squares, M2 with filled blue squares, and M7 with filled blue triangles, indicating the admixture with either "extreme" DW (red) or "moderate" DW (blue). Open black symbols in a) show the dSi range of the other mesocosm experiments as a comparison. a) dSi concentration (in μmol L-1), b) dSi drawdown (in
325 %) calculated by dividing the dSi concentration during the experiment with the initial dSi at the beginning of the experiment (day 1) and after replenishment of dSi on day 12. The horizontal line indicates the initial 100% calculated with the initial dSi on day 1 for the first 10 days and day 12 for the rest of the experiment. Data above the 100 % line must be influenced by mixing with high dSi from the BL c) N:P ratio (in M/M), d) N:Si (in M/M). N reflects all dissolved nitrogen species (DIN=NO<sub>x</sub> and NH<sup>4</sup>).

## 3.3 BSi concentrations and siliceous plankton development (diatoms and silicoflagellates)

At the beginning of the experiment, bSi concentrations in M1, M2, and M7 were similar, ranging from 3 to 4 μmol L<sup>-1</sup>, and mainly consisted of particles smaller than 20 μm, which accounted for 40 to 50% 335 of total bSi (Fig. S1). In M2 and M7 a strong increase in bSi was observed between days 3 and 4 (maximum bSi: 6.5 µmol L<sup>-1</sup>), whereas M1 remained constant over time (Table 1; Fig.3a). This was reflected by an increase in diatom cell counts obtained from microscopy data (see 2.3.; Table S1, Fig. 3b), which were highest during this time with abundances of 10 and 18 x 10<sup>6</sup> cells L<sup>-1</sup> in M7 and M2, respectively. When only considering silicifying plankton, diatoms contributed up to 100 % of the cells 340 counts (Fig. 3c). The most common diatom species were Skeletonema costatum, Cerataulina pelagica, Cylindrotheca closterium, Guinardia delicatula, Thalassiosira sp., Leptocylindrus danicus and Entomoneis alata var. alata (new name: Entomoneis paludosa). After day 8, bSi concentrations dropped to < 1 μmol L<sup>-1</sup> together with diatom cell counts below 1 x 10<sup>6</sup> cell L<sup>-1</sup>. Silicoflagellate abundances were generally low from day 1 to 10 in all mesocosms (< 0.006 x 10<sup>6</sup> cells L<sup>-1</sup>) and were 345 slightly increasing before DW addition on day 10 in M2 and M7 with up to 0.038 x 10<sup>6</sup> cells L<sup>-1</sup>. Two different silicoflagellate species were observed: Dictyocha fibula and Dictyocha octonaria (also known as Octactis octonaria; up to 30 µm in diameter without spines; for more details, see supplement Text S1). For a detailed list of all diatom and silicoflagellate cell counts, see supplement table S1.

After the DW addition (day 12), the bSi immediately increased, ranging between 2 μmol L<sup>-1</sup> 350 (M1) and 8 μmol L<sup>-1</sup> (M7) associated with higher silicoflagellate cell counts of up to 0.4 x 10<sup>6</sup> cells L<sup>-1</sup> on day 13. On day 15, bSi concentrations decreased to < 0.5 μmol L<sup>-1</sup> until day 35 and slightly increased towards the end of the experiment ranging between 1 and 2 μmol L<sup>-1</sup> (with a maximum of 4 μmol L<sup>-1</sup>). Overall, bSi closely correlated with diatom and silicoflagellate cell counts, except for day 20, when we observed the highest sillicoflagellate cell counts of 1.4 x 10<sup>6</sup> cells L<sup>-1</sup>. However, this does not affect the calculation of the fractionation factor as we used data from days 13 and 17.

Fig.3: a) bSi concentration (in μmol L<sup>-1</sup>) during the experiment. The DW additions on days 11 and 12 are marked by a grey bar. M1 data are indicated with open red squares, M2 with open blue squares and M7 with open blue circles indicating the admixture with either "extreme" DW (red) or "moderate DW (blue). Open black symbols in a) show the dSi range of the other mesocosm experiment as a comparison. b) diatom cell count (in x 10<sup>6</sup> cells per L<sup>-1</sup>). c) The fraction of siliceous plankton (diatoms and silicoflagellates; in %) with respect to all plankton (including non-siliceous organisms, like dinoflagellates) according to cell abundances from microscopic data. d) silicoflagellate cell counts (in cells per L<sup>-1</sup> x 10<sup>6</sup>). Please note the different scales in b) and d). in d), schematic figures of both silicoflagellate species are shown. The most abundant species is Dictyocha octonaria.

## 3.4 $\delta^{30}Si_{dSi}$ and $\delta^{30}Si_{bSi}$ in surface waters

δ<sup>30</sup>Si<sub>dSi</sub> signatures showed a large range from +0.75 ‰ to +4 ‰ during the 50-day long study period. During phase I, δ<sup>30</sup>Si<sub>dSi</sub> in all three mesocosms was similar, with values generally between +2 ‰ and +2.7 ‰ (Fig.4a). The DW was added on day 12 to the surface layer with a low δ<sup>30</sup>Si<sub>dSi</sub> signature of +1.5 ‰. Despite this relatively low δ<sup>30</sup>Si<sub>dSi</sub> signature of the DW, δ<sup>30</sup>Si<sub>dSi</sub> in all three mesocosms evolved to values ranging between +3 ‰ to +4 ‰ on day 14 (M7) and day 17 (M1, M2). This sharp increase in δ<sup>30</sup>Si<sub>dSi</sub> was caused by the rapid decline of dSi concentration (up to 50 % removal within 1 day, see also 3.2) shortly after DW addition. After day 17, δ<sup>30</sup>Si<sub>dSi</sub> values constantly decreased, with the lowest δ<sup>30</sup>Si<sub>dSi</sub> values observed in M1 and M2 at the end of the experiment (+0.75 to +1.2 ‰).

 $\delta^{30}Si_{bSi}$  was only analyzed in selected samples and ranged between +0.8 ‰ and +2 ‰. After the DW addition,  $\delta^{30}Si_{bSi}$  decreased to values between -0.5 ‰ and +0.6 ‰ and was then constantly increasing until day 24. The  $\delta^{30}Si_{bSi}$  signal was generally lower than that of  $\delta^{30}Si_{dSi}$ , with few exceptions at the end of the experiment (days 38 and day 50).

Fig. 4: a) Dissolved and particulate silicon isotope data per day. δ<sup>30</sup>Si<sub>dSi</sub> is marked by open symbols and δ<sup>30</sup>Si<sub>bSi</sub> by closed symbols. Red squares show data from M1, blue squares from M2, and blue squares from M7. The stars indicate δ<sup>30</sup>Si<sub>dSi</sub> of the "extreme" DW (red, +1.47 %) and the "moderate" DW (blue, +1.54 %). Both DW types show similar dSi concentrations with 17.43 and 19.60 μmol L<sup>-1</sup>. b).
Relative abundance of silicoflagellates (in %) obtained from cell counts compared to diatom abundances (M1: red-shaded area; M2: blue-shaded area with dashed line, and M7: blue-hatched area with solid line).

## 3.5 Isotopic fractionation

We observed a smaller offset between δ³0Si<sub>dSi</sub> and δ³0Si<sub>bSi</sub>, Δ³0Si<sub>bSi-dSi</sub>) within the first ten days of the experiment than during the days after DW addition. From day 1 to 10, Δ³0Si ranged between -0.2 % and -1.9 % (Fig. S2, Table 1). Directly after the DW addition, Δ³0Si decreased to -3.2 % (M1), -2.9 % (M2), and -2.6% (M7), respectively. After day 17, Δ³0Si markedly increased towards positive values (+0.1 % to +0.7%) at the end of the experiment (day 50).

The fractionation factor  $^{30}\epsilon_{mixed}$  was calculated after DW addition according to Equation (5) from three independent mesocosm experiments.  $^{30}\epsilon_{mixed}$  was -2.57± 0.38 ‰, -3.47 ± 0.42 ‰ and -3.60 ± 0.39 ‰ for M1, M2, and M7, respectively. Error estimates were derived from a Monte-Carlo simulation with 2000 random combinations of  $\delta^{30}\mathrm{Si}_{dSi}$  and  $\delta^{30}\mathrm{Si}_{bSi}$  values, taking a 2 s.d. error of 0.20 ‰ into account. To calculate the fractionation factor for silicoflagellates ( $^{30}\epsilon_{silicos}$ ), their relative abundance was taken into account and in M1, M2, and M7 was 62.3 %, 98.29 %, and 97.3 %, respectively. The resulting corrected  $^{30}\epsilon_{silicos}$  was -3.71 ‰, -3.51 ‰, and -3.67 ‰, with an average Si fractionation factor for silicoflagellates of -3.63 ±0.40 ‰ (2 s.d.; Table 2). The  $^{30}\epsilon_{silicos}$ , calculated with the minimum and maximum fractionation factor for diatoms (-0.5 ‰ and -2.1 ‰), gave -3.78 ± 0.22 ‰ and -3.55 ± 0.24 ‰, respectively, which are within error identical.

Table 2: Measured  $\delta^{30}Si_{dSi}$  and  $\delta^{30}Si_{bSi}$  values and the calculated fractionation factors ( $\delta^{30}\varepsilon_{mixed}$ ) for M1, M2, and M7.  $\delta^{30}\varepsilon_{Silicos}$  represents the corrected fractionation factor, taking the relative abundance of silicoflagellates into account. The error of the mean (2 s.d.) and the average error resulting from the Monte Carlo simulations (2 s.d. Monte Carlo) are given.

| Kosmos | Day 13<br>dSi | Day 13<br>δ³ºSi | Day 17<br>dSi | Day 17<br>δ³ºSi | f    | <sup>30</sup> ε (mixed) | Diatoms | Silicos | <sup>30</sup> ε (silicos) |
|--------|---------------|-----------------|---------------|-----------------|------|-------------------------|---------|---------|---------------------------|
|        | (initial)     | (dSi, initial)  | (final)       | (bSi)           |      | <b>‰</b>                | %       | %       | ‰                         |
| M1     | 8.91          | 2.12            | 7.65          | -0.26           | 0.86 | -2.57                   | 37.74   | 62.26   | -3.71                     |
| M2     | 8.90          | 2.93            | 5.91          | 0.12            | 0.66 | -3.47                   | 1.71    | 98.29   | -3.51                     |
| M7     | 5.97          | 3.40            | 3.56          | 0.65            | 0.60 | -3.60                   | 2.69    | 97.31   | -3.67                     |
|        |               |                 |               |                 |      |                         | Average |         | -3.63                     |
|        |               |                 |               |                 |      | 2 s.d.                  |         |         | 0.21                      |
|        |               |                 |               |                 |      | _                       | 0.40    |         |                           |

## 3.6. BSi increase per day, biovolume and biogenic silica content per cell

At the beginning of the experiment and after the addition of DW, a constant increase in bSi was observed over more than two sampling days in M2 and M7 (Fig. 3a). These data could be used to calculate the net increase in bSi per day (Fig. S3). Before the addition of DW, when mainly diatoms were abundant, the bSi increase per day was 1.1 and 1.2 μmol day<sup>-1</sup> for M2 and M7, respectively. After the DW addition, the bSi increase was 0.12, 1.31 and 2.3 μmol day<sup>-1</sup> for M1, M2 and M7, respectively, corresponding to the highest abundance of silicoflagellates (97%). Because a small number of very large diatom cells (>100 μm) could potentially influence the silicon cycle, we quantified the biovolume of the samples (biovolume normalized to cell counts). This analysis revealed that silicoflagellates were the dominant group after DW addition (Fig. S5). Overall, very few diatom cells ranging between 50 and 100 μm (e.g. Actinocyclus) were observed and no cell larger 100 μm (e.g. Coscinodiscus).

A very interesting finding was a pronounced difference in the relationship between bSi content and cell counts before and after DW addition. While we observed similar bSi concentrations (median

of 3.3 and 3.3 μmol L<sup>-1</sup>), the diatom-dominated phase (days 1-9) had cell counts several orders of magnitude higher (Fig. 5a), as here cells smaller than 20 μm were dominating. The bSi/cell ratio was significantly lower for the diatom (median: 1.2 pmol Si /cell; mean: 1.4 pmol Si /cell, min = 0.30, max = 4.6, n = 9) than for the silicoflagellate-dominated samples between day 13 and 20 (median: 17 pmol Si/cell, mean: 64.65 pmol Si/cell, min = 7.18, max = 181.20, n = 6, Fig. 5b). To minimize the bias introduced by a small number of disproportionately large cells, we highlighted samples containing 95% silicoflagellates. Within this subset, samples composed almost exclusively of silicoflagellates exhibited a median of 15.10 pmol cell<sup>-1</sup> (mean: 15.22 pmol cell<sup>-1</sup>). However, the estimated values must be treated with caution given that the error is particularly high for samples with low cell counts.

Fig. 5: a) Box plots showing the biogenic silica (bSi) concentration (left axis) before (day 1-9) and after DW addition (day13-19). Dots indicate higher cell counts (>10<sup>6</sup>) before DW water addition and lower cell counts after DW addition. b) Boxplot with included data points (black dots) reveal low pmol Si/cell values for the diatom-dominated phase (day 1-9) and higher pmol Si/cell values after DW addition (day

https://doi.org/10.5194/egusphere-2025-5079 Preprint. Discussion started: 28 October 2025 © Author(s) 2025. CC BY 4.0 License.

13-19), when silicoflagellates where dominating. Samples that contain more than 95 % diatoms are labelled in blue, samples with more than 95 % silicoflagellates in purble. Numbers next to the boxplots indicate the median. c) SEM picture of diatoms and Dictyocha fibula d) SEM picture of Dictyocha octonaria.

## 3.7 Comparison with Pacific water compositions outside of the mesocosms

In addition to the sampling within the mesocosms, we analyzed δ<sup>30</sup>Si<sub>dsi</sub> of Pacific seawater outside the mesocosms over the 50 days of the experiment. There, the dSi concentrations ranged between 4.7 and 16 μmol L<sup>-1</sup> with no apparent pattern (Fig. S4). The δ<sup>30</sup>Si<sub>dsi</sub> signature was highest within the first 5 days (+1.8 ‰), then remained constant near+1.6 ‰, possibly resulting from constant admixture of "new" dSi. The lowest δ<sup>30</sup>Si<sub>dsi</sub> value was observed on day 50 (+1.2 ‰, Table S1, Fig. S4). According to microscopic analysis, diatoms were the dominant phytoplankton (>99%), species throughout except for day 50, when diatoms only accounted for 30% of the silicifying community (Table S1).

### 455 4 Discussion

## 4.1. Phase I: diatom-dominated surface waters

The mesocosm study, conducted over a 50-day period with deep-water replenishment on day 12, provided insights into the evolution of δ<sup>30</sup>Si<sub>dSi</sub> and δ<sup>30</sup>Si<sub>bSi</sub> under closed-system conditions. During the initial 10 days, diatoms were the predominant phytoplankton group, making up nearly 100% of the siliceous phytoplankton community and up to 59% when including non-siliceous groups such as dinoflagellates (Bach et al., 2020). The diatom community was primarily composed of small, rapidly blooming species like *Skeletonema costatum* (less than 20 μm in diameter). Despite high DIN/dSi ratios (approximately 1 mol/mol) during this period (Fig. 2c), both chlorophyll α (not shown) and bSi concentrations showed a decline after day 1 in M1 and after day 4 in M2 and M7 (Fig. 3a; Table S1).

This reduction in productivity could be attributed to two main factors: a) significant light attenuation within the water column due to the high biomass standing stock in the surface layer (Bach et al., 2020), and b) the mixing with Chl α-depleted water of the bottom-layer (see method part 2.1).

During phase I, the  $\delta^{30}\mathrm{Si_{dSi}}$  signature remained stable (+2.1 to +2.5 ‰) indicating a balance between nutrient uptake (increase in  $\delta^{30}\mathrm{Si_{dSi}}$ ) and dissolution (decrease in  $\delta^{30}\mathrm{Si_{dSi}}$ ) in surface waters.  $\delta^{30}\mathrm{Si_{bSi}}$  (+0.4 ‰ to +1.5 ‰) was constantly lower than  $\delta^{30}\mathrm{Si_{dSi}}$  due to the "preferential" uptake of lighter isotopes leaving the seawater enriched in heavy isotopes (e.g., De La Rocha et al., 1997). This resulted in an apparent fractionation factor between silicifiers (mainly diatoms during the first 10 days) and seawater ( $\Delta^{30}\mathrm{Si}$ ) of around -1 ‰ ( $\epsilon^{30}$  value could not be estimated during the first 10 days, see section 2.5). This observation is in good agreement with laboratory studies investigating the Si isotope fractionation factor for diatoms with an overall mean  $\epsilon^{30}_{\mathrm{diatom}}$  of -1.1 ‰ (e.g., Sutton et al., 2013; 2018) as well as field estimates from the region ( $\Delta^{30}\mathrm{Si}$  of -0.4 ‰ to -1.1 ‰; e.g. Grasse et al., 2021).

### 4.1.2. Phase II: the dominance of silicoflagellates

Shortly before the addition of DW, diatom cell counts dropped and did not recover after the addition of DW despite high dSi concentrations (~6 to 9 µmol L<sup>-1</sup>) in the surface water (Fig. 2a). Instead, we observed an increase in the abundance of silicoflagellates, especially in M2 and M7, associated with higher bSi concentrations, although silicoflagellate cell counts were only 0.1 to 8% compared to diatom cell counts in Phase I (Fig. 5a). This abrupt shift in the phytoplankton community may have been caused by a strong decrease in DIN and therefore very low DIN:Si ratios (< 0.7). This creates unfavorable conditions for diatoms, which typically require a 1:1 DIN ratio for optimal growth (e.g. Brzezinski et al., 1985; 2008).

Silicoflagellates often occur together with diatoms, but compared to diatoms, they usually only play a subordinate role and occur less frequently, especially off the Peruvian Coast (e.g., Grasse et al., 2021; Ochoa et al., 2010). Under nutrient-rich conditions, diatoms typically outcompete silicoflagellates 490 due to their higher growth rates that can reach doubling times of less than 24 hours (Werner, 1970). In contrast, silicoflagellates replicate approximately every two days (Valkenburg & Norris, 1970). In seawater, the distribution and abundance of silicoflagellates in the water column may be influenced by a number of factors (e.g. water temperature, salinity, nutrient availability and grazing), but the relationship is neither simple nor consistent (e.g., Sancetta, 1990, for more details see Supplement S1). 495 Blooms are frequently observed in environments enriched in organic matter, which suggests that these organisms are capable of mixotrophy (Quéguiner, 2017). Mixotroph organisms can harness light energy to fix carbon dioxide while also obtaining organic carbon through the ingestion of prey, such as bacteria and other small protists. This dual nutritional mode is particularly advantageous in environments where nutrient availability fluctuates, such as during periods of low light or limited inorganic nutrient 500 availability.

In the upper ocean, dSi uptake and subsequent dissolution in subsurface waters are the main processes leading to a close correlation between dissolved seawater δ<sup>30</sup>Si<sub>dSi</sub> and Si concentration (Fig. 6). Since diatoms are the main phytoplankton species incorporating dSi, the isotopic signature thus generally serves as a tracer of the diatom-driven silicon cycle, with values ranging from +1.7‰ to +3.0‰ in the coastal surface ocean and about +1.5 ‰ in the subsurface (Ehlert et al., 2012, Grasse et al., 2021). Interestingly, the samples dominated by silicoflagellates deviated significantly from the above general trend in that higher δ<sup>30</sup>Si<sub>dSi</sub> were observed at moderate concentrations (5 to 9 μmol L<sup>-1</sup>), as a result of higher fractionation between seawater and silicoflagellates. The silicoflagellate bloom was associated with a pronounced increase in δ<sup>30</sup>Si<sub>dSi</sub> of up to +4.1‰ and low δ<sup>30</sup>Si<sub>bSi</sub> (-0.26 to + 0.65 ‰).

The increased fractionation between seawater and silicoflagellates (discussed in Section 4.2) leads to a deviation from the typical relationship between δ<sup>30</sup>Si<sub>dSi</sub> and dSi concentrations generally observed in the Peruvian OMZ (Fig. 6). This shift may explain some of the observed exceptionally high δ<sup>30</sup>Si<sub>dSi</sub> values (up to 4.4‰ in the Eastern Equatorial Pacific; Grasse et al., 2013) that deviate from the

commonly observed δ<sup>30</sup>Si<sub>dSi</sub>/ln(dSi) relationship). Future studies need to take into account that even low cell abundances of silicoflagellates can have a major influence on the δ<sup>30</sup>Si<sub>dSi</sub> signature in surface waters.

Fig. 6: Compilation of δ<sup>30</sup>Si<sub>dSi</sub> versus the natural logarithm of dSi displaying data of this study separated for samples from day 1 to 10 (open blue circles), when diatoms dominated the phytoplankton community and from day 13 to 20 (closed blue circles), when silicoflagellates were dominant. Colored dots indicate δ<sup>30</sup>Si<sub>dSi</sub> data from the Pacific from this study (sampled outside of the mesocosms; closed purple circles) and a previous cruise (open purple circles; M93) published in Grasse et al. (2021), where diatoms were the dominating phytoplankton. CS: closed system OS: open system. Regression lines for diatom-dominated samples are defined as 2.80 + (-0.39)\*x; r<sup>2</sup>=0.38 and for silicoflagellate-dominated samples: 5.82 + (-1.60)\*x; r<sup>2</sup>=0.78).

## 4.1.3. Phase III (days 44 to 50)

Towards the end of the experiment, a slight increase in bSi and a decrease in dSi was observed as a result of diatom growth after day 35. Orni-eutrophication during the last 10 days enabled new phytoplankton growth through the relief from N-limitation in the surface layer. Interestingly, δ<sup>30</sup>Si<sub>dSi</sub> only partly shows an increase expected from enhanced dSi utilization and also decreasing towards the end of the experiment with the lowest δ<sup>30</sup>Si<sub>dSi</sub> values observed during the KOSMOS experiment in M1 and M2 (+1.1 and +0.9 ‰, respectively). Relatively low δ<sup>30</sup>Si<sub>dSi</sub> values (minimum values of +1.2 ‰) are also observed in surface samples from the Pacific outside of the mesocosms (Fig. S4; Fig. 6: Pacific samples; comparison with all δ<sup>30</sup>Si<sub>dSi</sub> data from the region). Such low values have not yet been reported

for surface waters in the Peruvian Upwelling region. The lowest δ<sup>30</sup>Si<sub>dSi</sub> observed so far (+1.7 ‰, Ehlert et al., 2012) reflects recent upwelling with a low δ<sup>30</sup>Si<sub>dSi</sub> signature in subsurface waters (mean +1.5 ‰, Grasse et al., 2021; Fig. 5). Such values could instead be explained by lithogenic input of small particles characterized by low δ<sup>30</sup>Si<sub>dSi</sub> (-1 to -3 ‰; Sutton et al., 2018). Very small dust particles could only have been transported into the mesocosm by wind from the nearby island of San Lorenzo. The samples from the Pacific (outside the mesocosm) may also have been influenced by river runoff and enhanced lithogenic transport. During the sampling period, the Peruvian upwelling experienced a very rare coastal El Niño (Rodríguez-Morata et al., 2019), which caused unusually intense precipitation in Peru and exceptionally high freshwater runoff, resulting in elevated lithogenic inputs into coastal waters (Rodríguez-Morata et al., 2019), which is the most likely explanation for the observed low δ<sup>30</sup>Si<sub>dSi</sub> signatures.

### 4.2 The Si isotope fractionation factor of silicoflagellates

Our mesocosm experiment offered a unique opportunity to determine, for the first time, the silicon isotope fractionation factor of silicoflagellates—an organism group that has not previously been investigated in field-based silicon isotope studies and for which laboratory culturing remains particularly challenging (Taguchi and Laws, 1985). We calculated a mean  $^{30}$ Esilicos of -3.63  $\pm$  0.40 % (mean  $\pm 2$  s.d.) corrected for silicoflagellate abundances. This  $^{30}\varepsilon_{\text{silicos}}$  is higher than the fractionation 555 factor for diatoms that was determined during culture experiments (-1 %), but lower than the mean fractionation factor of sponges (-5 %) and choanoflagellates (-6.5 %), respectively (Hendry & Robinson, 2012; Marron et al., 2019; Sutton et al., 2018, Fig. 7). Up to now, the factors controlling Si fractionation in silicoflagellates have remained unclear, mainly due to numerous unknowns in their dSi uptake and silicification. The best-studied taxa with respect to silicification are diatoms. For diatoms, 560 it has been suggested that an important fractionation step may occur during cell uptake of dSi via specific sodium-coupled active silicate transporters (SITs, Hildebrand et al., 1997; Thamatrakoln et al., 2006). These studies have shown that at low dSi concentrations (below 30 μmol L<sup>-1</sup>), uptake is generally SIT-mediated, while at higher concentrations, dSi enters the cell by diffusion (Shrestha & Hildebrand, 2014; Thamatrakoln & Hildebrand, 2008). SITs have only been reported for most diatom species, 565 radiolaria, choanoflagellates and Florenciella sp., a non-siliceous stramenopile belonging to the dictyochophyte lineage ("silicoflagellates"). Therefore, transporter gene families in silicoflagellates containing a silicon skeleton remain unclear. A second transporter family important for dSi uptake in some phylogenetic lineages is the LSi2-like transporter found in choanoflagellates, sponges and diatoms (Marron et al., 2016). Due to the presence of different transporter families and the observation of 570 different fractionation factors between groups, several studies have suggested that there is a taxonomic component to Si isotope fractionation during biosilicification, possibly via a shared or related biochemical transport pathway (Hendry et al., 2018). Sponges and choanoflagellates belonging to the group "Opisthokonts" have high fractionation factors (up to -7 %; Hendry & Robinson, 2012; Marron

et al. 2019), whereas stramenopiles, e.g. diatoms, have been considered as a group with lower fractionation factors (Fig. S6). However, our new results revealing an intermediate fractionation factor for silicoflagellates challenge this taxonomic justification, as dictyochophytes ("silicoflagellates") also belong to the group of stramenopiles.

Figure 7: Overview of the fractionation factor for planktonic silicifiers (diatoms, silicoflagellates, 580 choanoflagellates). Radiolaria are not shown, as only an apparent fractionation factor, derived from core-top data (sediment) and seawater data, is available (Doering et al., 2021). Data is derived from De La Rocha et al. (1997), Sutton et al. (2013); Milligan et al. (2004); Meyerink et al. (2017), Sun et al. (2014); Marron et al. (2019) and this study.

Another possibility to explain the Si isotope fractionation in silicoflagellates would be that fractionation depends on other factors such as nutrient availability (e.g., dSi, Fe, NO<sub>3</sub><sup>-</sup>), which ultimately can affect Si uptake, growth rate, and bSi content in the cell (e.g. Paasche, 1973; Takeda, 1998). Calculated bSi/cell ratios were approximately 15 times higher than those of diatoms. The biogenic content of small, fast-blooming diatom cells is in good agreement with literature values, generally ranging between approximately 0.1 and 1 depending on light and nutrient limitation (e.g., Claquin et al., 2002). However, no published data for the bSi content of silicoflagellates are available for comparison. The elevated bSi/cell ratio indicates more strongly silicified cells, potentially with fewer pores which might be the reason for the strong fractionation. However, further research is required to clarify the processes driving the high fractionation factor.

Based on currently available knowledge, it remains unclear which process ultimately influences the degree of fractionation between seawater and silicifiers. Further studies, including culture studies of silicoflagellates and production rates, would be required for a better understanding.

δ<sup>30</sup>Si data obtained from siliceous phytoplankton (e.g., diatoms, radiolaria, and sponge spicules) in sediment cores have been used to gain insight into the mechanisms of the Si cycle of the past (e.g., 600 Doering et al., 2016a; Doering et al., 2016b; 2019; 2021; Hendry & Robinson, 2012). However, to reconstruct past dSi concentrations and utilization, it is necessary to know the fractionation factor during dSi uptake. Despite several studies that investigated silicoflagellate abundances in the past (e.g., Bukry, 1981; Amigo, 1999; McCartney, 2013;), no studies have been conducted so far on δ<sup>30</sup>Si signatures of silicoflagellates preserved in sediments. The obtained fractionation factor for silicoflagellates, therefore, provides the basis for a new paleo proxy for the reconstruction of the past Si cycle.

#### Conclusions

630

Our KOSMOS experiment off the coast of Peru provided a unique opportunity to study the silicon cycle of the Peruvian upwelling under controlled conditions. We tracked a shift in the phytoplankton 610 community from diatoms to silicoflagellates. The addition of nitrate depleted DW water (low DIN:Si ratios), which disadvantaged diatoms and favored the growth of mixotrophic silicoflagellates. Due to the low abundance of silicoflagellates in natural environments and the difficulties encountered when attempting to culture them, information on this taxonomic group is scarce. The silicoflagellate bloom was associated with a pronounced increase in  $\delta^{30}Si_{dSi}$  of up to +4.1 ‰ and low  $\delta^{30}Si_{dSi}$  (-0.26 to +0.65 615 %). Surface waters dominated by silicoflagellates exhibit a pronounced deviation from the characteristic relationship typically observed in the Peruvian OMZ between in  $\delta^{30}Si_{dSi}$  and dissolved dSi concentrations. We report, for the first time, a silicon isotope fractionation factor between seawater and silicoflagellates ( $^{30}\varepsilon_{\text{silicos}} = -3.6 \text{ }\%$ ). This value is significantly higher than the diatom fractionation factor ( $^{30}$ E<sub>diatoms</sub> = -1.1 ‰). The stronger fractionation expressed by silicoflagellates may be attributed 620 to their comparatively higher degree of silification. By determining a fractionation factor for silicoflagellates, we provide a novel tool for understanding dSi utilization in the past. Although silicoflagellates play a minor role in the ocean compared to diatoms, they can influence the silicon (Si) cycle, leading to increased silicon drawdown and bSi removal from the euphotic zone.

625 Data availability. The data will be published at PANGAEA.

Author contributions. UR, LTB, TB, MG designed the experiment. KD, TB, EvdE, AJP, AB, PG contributed to the sampling. PG, KD, AB, SSR and EvdE analyzed data. PG wrote the manuscript with comments from all co-authors.

Competing interests. At least one of the (co-)authors is a member of the editorial board of Biogeosciences.

https://doi.org/10.5194/egusphere-2025-5079 Preprint. Discussion started: 28 October 2025 © Author(s) 2025. CC BY 4.0 License.

## **Acknowledgements:**

This project was supported by the Collaborative Research Centre SFB 754 Climate-Biogeochemistry Interactions in the Tropical Ocean, financed by the German Research Foundation (DFG).

Additional funding was provided by the EU project AQUACOSM and the Leibniz Award 2012 granted to U.R. We thank all participants of the KOSMOS-Peru 2017 study for assisting in mesocosm sampling and maintenance and for the amazing team spirit during this campaign. We are particularly thankful to the staff of the Instituto del Mar del Perú (IMARPE) for their support during the planning, preparation and execution of this study and to the captains and crews of BAP MORALES, IMARPE VI and BIC HUMBOLDT for support during deployment and recovery of the mesocosms and various operations in the course of this investigation. Special thanks go to the Marina de Guerra del Perú, in particular the submarine section of the Navy of Callao, and to the Dirección General de Capitanías y Guardacostas. This work is a contribution in the framework of the Cooperation agreement between the IMARPE and GEOMAR through the German Ministry for Education and Research (BMBF) project ASLAEL 12-016 and the national project Integrated Study of the Upwelling System off Peru developed by the Direction of Oceanography and Climate Change of IMARPE, PPR 137 CONCYTEC.

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
