# Peer review of "Exploring Silicon Isotope Fractionation by Silicoflagellates: Results from a KOSMOS Experiment off Peru"

_EGUsphere, 2025_

## Author Comment (AC1)

We thank Jill Sutton for her thoughtful and constructive comments. Her comments are listed below, and our responses are provided in blue.

Grasse et al. provide the first *in situ* estimate of δ³⁰Si fractionation for silicoflagellates, which is a new and important contribution to understanding Si dynamics in the modern and past marine environments. The inclusion of the 2017 KOSMOS mesocosm experiment adds novelty by offering a controlled environment to study Si isotope fractionation, a major limitation in field-based studies. While the manuscript is well written and informative, several sections would benefit from clearer articulation of the scientific objectives, improved structure, and a more detailed explanation of uncertainties. In addition, some broader implications were underdeveloped given that their short summary indicates that this information would be "providing a novel tool for understanding dSi utilization in the past. The paper could strengthened by more explicitly connecting the results to: (1) global δ³⁰Si budgets, (2) paleoceanographic reconstructions (3) potential biases in interpreting δ³⁰Si (water column and sediment). Below is a list of comments suggesting both major and minor revisions.

1) Global δ³⁰Si budget: A major uncertainty in evaluating the influence of silicoflagellates and their higher Si isotope fractionation factor comes from the fact that we lack reliable global estimates of silicoflagellates abundance in the modern ocean. Most authors cite Riedel (1959), who reported that silicoflagellates skeletons typically account for only 1-2% of the siliceous fraction in marine sediments. While this provides a useful sedimentary benchmark, it does not necessarily reflect their standing stock or production in the water column, as their abundances can be very patchy, seasonally variable, and regionally restricted. Without better observational data, it is challenging to determine whether their higher fractionation factors could meaningfully bias large-scale silicon isotope estimates. However, while we do not expect silicoflagellates to exert a major influence on the δ³⁰Si budget in the modern ocean, the situation may have been different in the geological past. Several studies indicate that silicoflagellates experienced periods of much higher abundance, particularly during the Late Cretaceous to early Paleogene. During these intervals their contribution to biogenic silica production may have been large enough to influence local or even basin-scale silicon-isotope signatures. While we can only hypothetical assumptions about their potential impact, dedicated studies would be required to properly evaluate this possibility.

Point 2) and 3) are discussed in detail below. The new information will be added to the discussion.

Major comments:

The introduction provides extensive background on the Peruvian Upwelling, Si cycling dynamics, plankton community dynamics and Si isotopic fractionation, but the primary research questions or hypotheses of the study should be more clearly presented. For example, the central scientific question is a little bit buried: "What is the Si isotope fractionation factor for silicoflagellates, and how does it influence δ³⁰Si in dynamic upwelling systems?". This should be explicitly stated towards the end of the Introduction.

We will add further information to the last paragraph of the introduction.

*"In 2017, we conducted a KOSMOS (Kiel Off-Shore mesocosms for Future Ocean Simulations) experiment off Lima (Peru) for 50 days to simulate upwelling with different nutrient stoichiometries*

*(N:Si and N:P). While upwelling of nutrient rich water in the Peruvian upwelling generally induce diatom blooms (e.g. Franz et al., 2012, Grasse et al., 2021), low N:Si ratios would lead to shifts in the phytoplankton communities. The unique setting of the experiment made it possible to study the evolution of δ³⁰Si$_{dSi}$ and δ³⁰Si$_{bSi}$ in a closed system without the influence of horizontal water mass mixing. After the addition of nutrient depleted deep water, we observed a shift from a diatom-dominated community (day1-10, phase I) to a (silico-)flagellate-dominated community (day 13 to 42, phase II). This shift provided novel insights into the silicon cycle in the Peruvian upwelling system and enabled, for the first time, the determination of a silicon isotope fractionation factor for silicoflagellates. Importantly, these data also allow, for the first time, an assessment of the potential influence of silicoflagellates on the dissolved silicon isotope composition in upwelling regions, and thus under which environmental conditions their contribution may exert a measurable effect on the marine Si isotope distribution.*

While diatom dynamics are thoroughly explained in the introduction, a discussion of (silico) flagellates is somewhat brief and lacks detail. Given that the study presents the first Si isotope fractionation factor for silicoflagellates, more ecological and physiological context (and references) would be useful. For example, lines 69-73 lack references and there appears to be an error in the paragraph. Some questions to address include: (1) How do silicoflagellate silica structures differ from diatom frustules? (2) What is known about their silica uptake pathways? The emphasis of what is unknown will help the reader understand the importance of the questions being addressed.

The sentence (former L69-73) was corrected as follows: "These shifts in phytoplankton composition strongly affect the marine silicon cycle, but also the carbon cycle. While carbon uptake rates in some silicoflagellate species (e.g. *Dictyocha perlaevis*) have been shown to be comparable to other phototrophic phytoplankton species (e.g. Taguchi & Laws 1985), there are still insufficient data on carbon uptake differences between different species of silicoflagellates or changes caused by the variability of environmental conditions (Closset et al. 2025)."

We also added further information to the introduction:

1) How do silicoflagellate silica structures differ from diatom frustules?

   While diatoms have two overlapping valves, the silica skeleton of silicoflagellates is an external, intricate, and rigid framework of hollow, opaline silica tubes, forming a basket-like structure (Preisig 1994). For few species it was possible to show that they can have a double skeleton (sometimes interpreted as pre-division stages). The paired skeletons are not mirror images but share the same rotational orientation, forming dome-shaped halves of a more spherical structure, a pattern with important implications for skeletal function, phylogeny, and the generic distinction of living and fossil silicoflagellates (McCartney et al. 2014).

2) What is known about their silica uptake pathways?

To our knowledge there are no studies on the silica uptake pathways in silicoflagellates

Also, I found that the presentation of the KOSMOS mesocosm experiment in the introduction lacked some detail. A clearer statement of the study targets (i.e. specific processes) and why these findings mater for broader oceanographic or paleoceanographic applications should be highlighted.

The KOSMOS study investigated the effects of upwelling of deep waters with varying N:P and N:Si ratios, and how these ratios influence the composition of phytoplankton and zooplankton. However,

the induced upwelling exhibited lower N:P and N:Si ratios than anticipated, leading to unexpected conditions such as a silicoflagellate bloom.

Silicoflagellates are often overlooked due to their lower abundance compared to diatoms. This lower abundance in surface waters is reflected in the sediments below: While they are present in several of the shorter sediment cores from the Peruvian upwelling (personal communication Kristin Doering), they likely play a dominant role for biogeochemical conditions under which diatoms are absent. Their occurrence following diatom blooms may also help constrain the role of non-diatom silicifiers in the silicon cycle, and knowledge of their fractionation factor can potentially improve estimates of $\delta^{30}Si$ signatures of dissolved silica under nutrient-limiting conditions, allowing an assessment of whether DSi was fully consumed or whether nitrate limitation prevailed.

We will add further clarification to the last part of the discussion.

While nutrient supply mechanisms (vertical advection, mixing, seasonal variability) are described accurately, the manuscript somewhat underplays mesoscale variability, differences in water mass sources, and timescale interactions (physical vs. biological). All of these can strongly shape local $\delta^{30}Si$ signatures and community composition. Briefly incorporating these factors would produce a more comprehensive discussion of why field-based $\delta^{30}Si$ measurements are difficult to interpret.

We agree and will incorporate a sentence that highlights the complexity of temporal and spatial evolution of hydrographic conditions and changes in physical and biological interactions and their importance for the resulting dissolved and particulate $\delta^{30}Si$ signal. The following sentence was added to the discussion L516:

*"Overall, the interpretation of silicon isotopes in field studies is complicated, not only due advection of differencent water masses (horizontally and vertically), but also due to processes on different time scales. While seawater can integrate a signal of Si utilization as well as dissolution on rather short time scales, the particles reflect the evolution of the signal over longer time scales."*

The critique of Rayleigh and steady-state models is important, but the manuscript does not fully explain how these limitations influence isotopic interpretations in practice. The mesocosms are semi-closed, but episodic mixing (bottom layer intrusion, biomass movement) introduces potential non-Rayleigh effects. This should be more thoroughly addressed. Specifically: (1) Do oversimplified $\delta^{30}Si$ assumptions bias estimates of Si utilization? (2) Are there specific examples from past studies where these biases have been demonstrated and/or discussed?

Vertical mixing in the mesocosms was minimized by adding saltwater to the bottom layers on days 13 and 33 (see Bach et al., 2020). Mixing between surface waters and the deep-water additions mainly occurred prior to the additions and around day 30; therefore, the days for which the Rayleigh fractionation model was applied to estimate the silicoflagellate fractionation factor experienced only minimal vertical mixing. Although the Rayleigh model likely provides the best estimate under these conditions, we now clarify in the text, that the surface layer was not a completely closed system given that minor mixing events, dissolution, and biomass sinking may have influenced surface water $\delta^{30}Si$ values. Previous field studies in the Peruvian upwelling have shown that the Rayleigh-type fractionation model often underestimates the fractionation factor of diatoms (Ehlert et al. 2013; Grasse et al. 2021), due to upwelling and effects of dissolution.

As Kosmos 1 contains a larger fraction of diatoms, which have growth rates different from silicoflagellates (see comments reviewer 1), we now only use Kosmos 2 and 7 for the calculation of the Si isotope fractionation factor of silicoflagellates.

While we are confident that the fractionation factor of silicoflagellates is significantly higher than that of the investigated diatom species, we will add additional context to the manuscript clarifying uncertainties and that further studies, particularly culture experiments, are needed to verify the silicoflagellate fractionation factor.

Minor comments:

For the silicoflagellate ε calculation, the assumption that silicoflagellates dominate uptake between days 13–17 is reasonable but requires more explicit demonstration (perhaps with size-fraction bSi?). Also, is there a possible role of dissolution (diatom or other organisms) in this zone? How would this influence the interpretation of the ε30Si?

We will include further information on the diatom species, size distribution as well as biovolume specifically for day 13 and 17.

Dissolution likely occurs throughout and would tend to lower the apparent fractionation factor for silicoflagellates. This effect is illustrated in Figure 6, where $\delta^{30}Si$ is plotted against ln(dSi), a relationship governed by the balance between silicate utilization and dissolution, with the slope reflecting the net fractionation. Fully constraining the magnitude and role of dissolution, particularly for silicoflagellates, requires further investigation, which is beyond the scope of this study.

We will add a paragraph to the Discussion section addressing potential processes that may influence the estimated silicon isotope fractionation factor of silicoflagellates.

Line 106 - "admixture from multiple sources" Do you mean "a mixture.."?

Will be corrected

Lines 84-92 - Also, see Frings et al. 2024 for newer information.
.https://doi.org/10.1016/j.quascirev.2024.108966

We searched for additional information in Frings et al., 2024 and will add a sentence in L90:

*"Further experiments suggest that isotopic fractionation is primarily driven by early kinetic effects during rapid silica precipitation rather than biomolecule-specific processes (Cacarino et al., 2021)."*

Line 540 – perhaps cite Cotard et al. 2025 (https://doi.org/10.1002/lno.70243) for the lithogenic input ? They have some interesting evidence supporting that lithogenic input could affect the dSi composition. Is there any other evidence that could support this argument? For instance, rainfall, sediment load, turbidity, wind direction, proximity to dust sources?

We agree and will modify the sentence accordingly:

*"Such values could instead be explained by input of small lithogenic particles characterized by low $\delta^{30}Si_{dSi}$ (-1 to -3 ‰; Sutton et al., 2018). Even lower values (-0.33 ‰) in surface waters were observed above the Northern Kerguelen Plateau, clearly influenced by lithogenic input (Cotard et al., 2025)."*

As we described in the manuscript, the Coastal El Niño affected the region around the mesocosms through heavy rainfall and enhanced sediment load from the rivers (e.g. Rodríguez-Morata et al. 2019; Geilert et al. 2023). Unfortunately, the additional datasets provide no further insight into

potential lithogenic inputs. The trace-metal measurements (e.g., Fe, Cd, Zn) varied widely with no clear pattern, most likely caused by contamination within the mesocosms (pers. communication Mark Hopwood). Likewise, the PAR sensor data offer no evidence for enhanced particle loads or light attenuation that would indicate lithogenic material entering the system.

We know from personal communication that the wind pattern had shifted at the end of the experiment, which transported very small dust particles. We plotted the wind direction and speed for day 40 to 47 (beginning of Apil 2017) and the last days of the experiment (day 48 and 50) with ERA5 hourly data on single levels from 1940 to the present (https://cds.climate.copernicus.eu/datasets/reanalysis-era5-single-levels?tab=overview) Unfortunately, the resolution is too low to show the local wind situation (Fig. R1, Experiment location marked by grey box). Wind data from the Isla Lorenzo weather station near Lima were available only as monthly averages and therefore do not capture short-term wind variability (Fig. R2).

[Figure]

*Fig. R1: Wind direction (arrows) and speed (color) for the study area before small dust particles were transported inside the mesocosm (day 40-47) and thereafter (da 48 -50). The grey square indicates the location of the KOSMOS Experiment.*

[Figure]

*Fig. R2: Wind direction and speed (m/s) data from a weather station on Isla San Lorenzo (Close to Lima) are only available monthly from January until April 2017 (Data source: Meteoblue). The last two sampling days of the experiment are marked by a red bar (14th and 16th of April).*

**References**

Closset, I., Baronas, J. J., Torricella, F., Tombeur, F. de, Liguori, B. T. P., Petrucciani, A., Bryan, N., López-Acosta, M., Churakova, Y., Thielecke, A. U., Zhang, Z., Monferrer, N. L., Pickering, R. A., Guyomard, M., & Zhu, D. (2025). Silicification in the ocean: from molecular pathways to silicifiers' ecology and biogeochemical cycles. *Ocean Science*, *21*(6), 3427–3470. https://doi.org/10.5194/os-21-3427-2025

Ehlert, C., Grasse, P., & Frank, M. (2013). Changes in silicate utilisation and upwelling intensity off Peru since the Last Glacial Maximum – insights from silicon and neodymium isotopes. *Quaternary Science Reviews*, *72*, 18–35. https://doi.org/10.1016/j.quascirev.2013.04.013

Geilert, S., Frick, D. A., Garbe-Schönberg, D., Scholz, F., Sommer, S., Grasse, P., Vogt, C., & Dale, A. W. (2023). Coastal El Niño triggers rapid marine silicate alteration on the seafloor. *Nature Communications*, *14*(1), 1676. https://doi.org/10.1038/s41467-023-37186-5

Grasse, P., Haynert, K., Doering, K., Geilert, S., Jones, J. L., Brzezinski, M. A., & Frank, M. (2021). Controls on the Silicon Isotope Composition of Diatoms in the Peruvian Upwelling. *Frontiers in Marine Science*, *8*, 697400. https://doi.org/10.3389/fmars.2021.697400

Ireland, H. A., & Riedel, W. R. (1959). *Silica in Sediments*. https://doi.org/10.2110/pec.59.01.0080

McCartney, K., Witkowski, J., Jordan, R. W., Daugbjerg, N., Malinverno, E., Wezel, R. van, Kano, H., Abe, K., Scott, F., Schweizer, M., Young, J. R., Hallegraeff, G. M., & Shiozawa, A. (2014). Fine structure of silicoflagellate double skeletons. *Marine Micropaleontology*, *113*, 10–19. https://doi.org/10.1016/j.marmicro.2014.08.006

Preisig, H. R. (1994). Siliceous structures and silicification in flagellated protists. *Protoplasma*, *181*(1–4), 29–42. https://doi.org/10.1007/bf01666387

Rodríguez-Morata, C., Díaz, H. F., Ballesteros-Canovas, J. A., Rohrer, M., & Stoffel, M. (2019). The anomalous 2017 coastal El Niño event in Peru. *Climate Dynamics*, *52*(9–10), 5605–5622. https://doi.org/10.1007/s00382-018-4466-y

Taguchi, S., & Laws, E. (1985). Application of a single-cell isolation technique to studies of carbon assimilation by the subtropical silicoflagellate Dictyocha perlae vis. *Marine Ecology Progress Series*, *23*, 251–255. https://doi.org/10.3354/meps023251

---

## Author Comment (AC2)

We thank the reviewer for their thoughtful and constructive comments. The reviewers' comments are listed below, and our responses are provided in blue.

Grasse et al investigated the Si processes regulated by diatoms and silicoflagellates by pelagic mesocosms in coastal Peru upwelling areas and stable Si isotopes. Overall, this is an interesting study and provide important knowledge of Si isotope fractionation of -3.63 ‰ during silicoflagellate production, implying a potential important application when investigating sedimentary records of biogenic Si. Prior to its publication, I have the following major and minor comments.

Major comments:

Regarding the calculation of Si isotope fractionation factor for silicoflagellates, the authors are missing an in-depth assessment of uncertainties. Given the data, diatoms and silicoflagellates are growing at different rates and exhibiting different abundance over the entire period. During the first 10 days, the production is exclusively dominated by diatoms, while thereafter, silicoflagellates take over within days. This indicates the growth rate of silicoflagellates is higher than diatoms and meanwhile consume more DSi relative to per unit cell of diatoms (this seems possible to be estimated given the experimental data), which in turn raise a question of what is a proper number for the initial d30Si value of DSi for them to grow. It is fine to use Eq 6 to estimate a mixed 30ε value, but this is probably only valid when assuming both diatoms and silicoflagellates keep their growth rate constant and the difference between these two growth rates remain the same. This means, even if they can share the same d30Siinit value derived from DW, the fraction of DSi, f, could be varied at each time point for diatoms and silicoflagellates. Subsequently, Eq 7 calculates the 30ε for silicoflagellates using relative abundances, but since their Si/cell differ much as shown in Fig 5b, should this be taken into account in the f in Eq 7?
Perhaps I make this question overcomplicated, but another thought is M2 and M7 is dominated by silicoflagellates by over 97% in certain days, why we need to bother the mixture with diatoms? It could be straightforward to use those data in those days to directly calculation 30ε for silicoflagellates.
I may miss some important information in this, but since this is the fundamental part and the main conclusion of the ms, such uncertainties should be clarified before publication.

We agree, that the simplified equation may not capture all the factors influencing the fractionation factor, such as growth rate or bSi content per cell. However, adding additional parameters to the equation would even induce more uncertainties, as growth rates as well as BSi content per cell vary not only between diatoms and silicoflagellates, but also between different diatom species.

We recalculated the fractionation factor for Kosmos 1 (a mixture containing diatoms and silicoflagellates) using biovolume data from Olenia et al. (2015) instead of cell counts for both groups on Day 17. It should be noted that phytoplankton species can vary greatly in cell size, as reflected in the biovolume, and not all cell counts refer to specific taxa (e.g. listed as pennates or only the genus), which leads to further uncertainties. Using biovolume instead of cell counts, the fractionation factor for M1 is lower ( -2.7 ‰) compared to the previous estimate (-3.71 ‰) as the biovolume for diatoms only accounts for 9 % compared to 91 % for silicoflagellates (cell counts were: 37.74 % diatoms, 62.27 % silicoflagellates)

We agree with the reviewer that estimating the Si isotope fractionation factor in experiments with mixed diatom and silicoflagellate assemblages is more complex and potentially biased. We therefore restricted our analysis to fractionation factors from two independent experiments (Kosmos 2 and Kosmos 7), where silicoflagellates comprised up to 99% of the biovolume and were thus consistent

with the corresponding cell counts (97–98%). The new mean fractionation factor for silicoflagellates is −3.54 ‰, which is, however, within the uncertainty range of the previously estimated value of −3.63 ‰.

While we are confident that the fractionation factor for silicoflagellates is significantly higher than that of the investigated diatom species, we will add additional context to the manuscript clarifying uncertainties and that further studies, particularly culture experiments, are needed to verify the Si isotope fractionation factor for silicoflagellates.

The table and text will be adjusted accordingly. Equation 7 will be removed from the manuscript. The revised Table 2 is presented below. The mean biovolume data will be added to the main text.

| Kosmos | Day 13 | Day 13 | Day 17 | Day 17 | $f$ | $^{30}\varepsilon$(Silicos) |
|---|---|---|---|---|---|---|
| | dSi (initial) | $\delta^{30}Si$ (dSi, initial) | dSi (final) | $\delta^{30}Si$ (bSi) | | ‰ |
| M2 | 8.90 | 2.93 | 5.91 | 0.12 | 0.66 | -3.47 |
| M7 | 5.97 | 3.40 | 3.56 | 0.65 | 0.60 | -3.60 |
| | | | | Average | | -3.54 |
| | | | | 2 s.d. | | 0.18 |
| | | | | (2 s.d Monte Carlo) | | 0.40 |

Other suggestions:

For clarity, it is better to say Si isotope fractionation factor in the text, or at least isotope fractionation factor, instead of "fractionation factor".

We agree and will adjust this accordingly throughout the text.

Line 47 the superscript "-" should be removed.

Will be removed

Line 70 "dinoflagellates" should start with the captial letters.

We will correct the sentence and add further information.

*"These shifts in phytoplankton composition strongly affect the marine silicon cycle, but also the carbon cycle. While carbon uptake rates in some silicoflagellate species (e.g. Dictyocha perlaevis) have been shown to be comparable to other phototrophic phytoplankton species (e.g. Taguchi & Laws 1985), there is still insufficient data on their carbon uptake to allow comparison of different species of silicoflagellates or changes in environmental conditions (Closset et al. 2025)."*

Line 184 "at" should be removed.

Accepted

Line 173-174 vs Line 207-209 Why BSi was digested at different NaOH concentration and temperature? Are there any specific reasons? And for BSi contents and its Si isotope measurement, how the authors assess the contribution from non-biogenic Si particles?

There are several protocols for determination of bSi concentrations with slightly different NaOH concentrations, temperatures. The bSi concentration during the mesocosm experiment was determined by the Mesocosm Team, which applied a method involving 0.1 NaOH at 85°C (see section 2.2). We used a protocol adapted from Mark Brzezinski's laboratory (UC Santa Barbara) involving 0.2 N NaOH in a 90 °C water bath to prepare samples for Si isotope measurements (please note that the text stated 95 °C, which will be corrected). In a second leach step, the filter was treated with 0.5 ml of 2.5 M HF for 48 hours to dissolve lithogenic material. To determine the optimal digestion times for silicon isotope measurements, we used test filters from different days with varying diatom and silicoflagellate abundances. Tests were conducted for a maximum of 150 minutes. A steep increase in the bSi content signal was observed during the first 80 minutes, given that bSi dissolves faster than lithogenic material. No significant increase in bSi was detectable thereafter (less than 5%), except for 2 samples from Day 1, which contain more lithogenic Si (up to 18%). This will be noted in the manuscript, and the corresponding data points will be highlighted in Figure 4.

Although Ragueneau and Tréguer (1994) pointed out that up to 15% of lithogenic silicate (LSi) can dissolve during sodium hydroxide digestion, our own measurements provide evidence that LSi dissolution in our samples, especially the samples used to determine the fractionation factor, was much lower. Assuming that the samples contained up to 15% LSi, the reported $\delta^{30}$Si values may be underestimated by 0.2‰, assuming a mean isotope signature of −1.07‰ for clay minerals. Lithogenic primary minerals are heavier at −0.2‰ (Sutton et al., 2018), resulting in an offset of 0.02‰. Both values are within the analytical error margin. We did not measure Al/Si ratios in the bSi samples as these may have been heavily biased within the mesocosms, which are not trace metal-free. Secondly, as we demonstrated in Grasse et al. (2021), the correction method with Al has its limitations, as the Al/Si ratios in bSi samples depend on external factors and the conditions of the diatom cells (living versus dead) and do not exclusively indicate contamination with lithogenic material.

However, we would like to point out, that bSi samples, which were used to determine the fractionation factor (day 17) only contained negligible amount of LSi (3%).

We will add this information to the manuscript.

Line 315 and some relevant text in the result section. What is the application of DIP in this study? Are they just used to show DIP is not a limiting nutrient for primary producivity?

The information on dissolved inorganic phosphate (DIP) was added to the results section, as it is a relevant parameter during the mesocosm experiments. We therefore would like to keep Figure 2d and add further content to the manuscript.

Line 600-605 I agree that we should be more careful when using sedimentary bSi to reconstruct dSi utilization. But this is also dependent on the relative abundance of each bSi species in sediment records of the studied area and the purification of these species for isotope analysis.

We agree with the reviewer and will add additional information to the last section of the main discussion part:

"$\delta^{30}$Si data obtained from siliceous phytoplankton (e.g., diatoms, radiolaria, and sponge spicules) in sediment cores have been used to gain insight into the mechanisms controlling the Si cycle of the past (e.g., Doering et al., 2016a; Doering et al., 2016b; 2019; 2021; Hendry & Robinson, 2012). However, reconstructing past dSi concentrations and utilization requires knowledge of species abundances in the sediment, in addition to careful purification of the samples used for Si isotope analysis. Further insights can be obtained from the fractionation factor associated with dSi uptake. Despite several studies that

*investigated silicoflagellate abundances in the past (e.g., Bukry, 1981; Amigo, 1999; McCartney, 2013;), no studies have been conducted so far on $\delta^{30}$Si signatures of silicoflagellates preserved in sediments. The obtained fractionation factor for silicoflagellates, therefore, provides the basis for the establishment of a new paleo proxy for the reconstruction of the Si cycle of the past."*